# Laparoscopic Pancreatoduodenectomy in Elderly Patients: A Systematic Review and Meta-Analysis

**DOI:** 10.3390/life12111810

**Published:** 2022-11-07

**Authors:** Adrian Bartos, Simona Mărgărit, Horea Bocse, Iulia Krisboi, Ioana Iancu, Caius Breazu, Patricia Plesa-Furda, Sandu Brînzilă, Daniel Leucuta, Cornel Iancu, Cosmin Puia, Nadim Al Hajjar, Lidia Ciobanu

**Affiliations:** 1Medicine Faculty, Iuliu Hațieganu University of Medicine and Pharmacy, 400012 Cluj-Napoca, Romania; 2Prof. Octavian Fodor Regional Institute of Gastroenterology and Hepatology, 400012 Cluj-Napoca, Romania

**Keywords:** elderly, pancreaticoduodenectomy, laparoscopic pancreaticoduodenectomy, mortality rates

## Abstract

Background and Aims: Recent single-center retrospective studies have focused on laparoscopic pancreatoduodenectomy (LPD) in elderly patients, and compared the outcomes between the laparoscopic and open approaches. Our study aimed to determine the outcomes of LPD in the elderly patients, by performing a systematic review and a meta-analysis of relevant studies. Methods: A comprehensive literature review was conducted utilizing the Embase, Medline, PubMed, Scopus and Cochrane databases to identify all studies that compared laparoscopic vs. open approach for pancreatoduodenectomy (PD). Results: Five retrospective studies were included in the final analysis. Overall, 90-day mortality rates were significantly decreased after LPD in elderly patients compared with open approaches (RR = 0.56; 95%CI: 0.32–0.96; *p* = 0.037, I^2^ = 0%). The laparoscopic approach had similar mortality rate at 30-day, readmission rate in hospital, Clavien–Dindo complications, pancreatic fistula grade B/C, complete resection rate, reoperation for complications and blood loss as the open approach. Additionally, comparing with younger patients (<70 years old), no significant differences were seen in elderly cohort patients regarding mortality rate at 90 days, readmission rate to hospital, and complication rate. Conclusions: Based on our meta-analysis, we identify that LPD in elderly is a safe procedure, with significantly lower 90-day mortality rates when compared with the open approach. Our results should be considered with caution, considering the retrospective analyses of the included studies; larger prospective studies are required.

## 1. Introduction

Laparoscopic surgery has become the gold standard treatment for many surgical conditions previously treated by the open approach [1,2,3]. As world’s population is aging and the life expectancy is rising, it seems that laparoscopy might respond to the surgical needs of the elderly [4].

Recent advances in the surgical techniques offer better outcomes for the patients, increasing the safety of the procedures; however, thorough understanding of surgical technique and the comorbidities by which elderly patients suffer is crucial for any surgeon [4].

Within open surgeries, pancreatoduodenectomy (PD), also known as the Whipple procedure, is considered the most complex procedure for a general surgeon, requiring the highest level of surgical expertise. Its history starts from the 19th century with ominous prohibitive mortality, which is now reduced to less than 2% [5].

Laparoscopic PD (LPD) was first described by Gagner and Pomp in 1994 [6]. However, there has not been a wide acceptance of this approach to date. Both hybrid approaches with conversion to open surgery and total LPD (TLPD) have been described [7,8,9]. Laparoscopic surgery is intended to reduce the systemic inflammatory response that is usually seen after open surgery; the systemic inflammatory response is responsible for postoperative complications. However, one randomized controlled trial (RCT) reported that the LPD, when compared with open PD (OPD), did not reduce the postoperative inflammatory response; IL-6 levels were associated with postoperative complications and pancreatic fistula [10].

Relevant literature comparing the LPD versus OPD are still limited, as revealed by the Evidence Map of Pancreatic Surgery [11,12], a continuous updated systematic review with meta-analyses offered by the International Study Group of Pancreatic Surgery (ISGPS). Three RCTs reported that in highly experienced hands, LPD was a safe and feasible procedure [13,14,15] that was associated with a significantly shorter length of stay. Conflicting results were noticed regarding the short-term morbidity and mortality rates in RCTs comparing LPD with OPD. Some authors reported that the patients with LPD had similar rates of 90-day mortality and of serious postoperative morbidities as those with open approach [13,15]. However, the RCTs conducted by van Hilst [16] were prematurely terminated because of a difference in 90-day complication-related mortality (risk ratio = 4.9, 95%CI 0.6–40.4). In the RCT conducted by Poves et al. [14], patients with LPD experienced significantly lower severe complication than those with OPD (Clavien–Dindo > 3).

Integrating all these results, we might assume that the clinical benefit of LPD compared with OPD might be minor despite extensive procedural expertise. A meta-analysis of three RCTs including 224 patients reported no significant benefit of laparoscopic approach compared to OPD in all aged categories [17]; no significant differences were noted regarding the 90-day mortality and perioperative morbidity. In this meta-analysis, the limitation included high risk of bias and moderate to very low certainty of evidence.

Future research is needed to identify the populations that will benefit from LPD [13]. One beneficiary category from minimally invasive technique might be represented by elderly patients. However, in elderly patients associating cardiovascular or pulmonary comorbidities, the benefits of minimally invasive dissection techniques might be outweighed by the risks (acidosis, changes in pulmonary mechanics, alteration in hemodynamic function and aspiration) [4].

In elderly patients, there is still a lack of evidence from RCTs to conclude the superiority of the minimally invasive technique for PD. Over the past decade, some single-center retrospective studies have focused on the results of LPD in elderly patients [18,19,20,21,22]. In these studies, two approaches were considered: (1) comparison between the laparoscopic and open approaches in the elderly; (2) comparison between the elderly and non-elderly population after LPD. In this current study, we combine the results from existing literature [18,19,20,21,22] to investigate the efficacy and safety of LPD in elderly patients by comparing with OPD.

The first representative study focusing on elderly patients was conducted by Tee et al. [21], who reported its retrospective analyses in 2015. In elderly patients, the incidences of cardiac events (OR 3.21, *p* < 0.001), respiratory events (OR 1.68, *p* = 0.04), delayed gastric emptying (OR 1.73, *p* = 0.003), increased length of stay (*p* < 0.001), discharge disposition other than home (OR 8.14, *p* < 0.001) and blood transfusion (OR 1.48, *p* = 0.05) were greater than in non-elderly patients, the morbidity and mortality did not differ between the OPD and LPD groups of elderly patients.

In the largest study, involving 1768 patients aged ≥ 75 years who underwent LPD (248, 14.0%) or OPD (520, 86.0%), Chapman et al. [20] reported better outcomes in LPD vs. OPD; 90-day mortality was significantly lower (7.2 vs. 12.2%, *p* = 0.049) in the LPD group. Additionally, median overall survival was significantly longer in the LPD group (19.8 vs. 15.6 months, *p* = 0.022). After adjusting for patient and tumor-related characteristics, there was a trend towards improved survival in the LPD group (HR 0.85, 95%CI 0.69–1.03).

Shin et al. [22] collected data on elderly patients aged ≥70 years (56 patients allocated to the LPD group and 270 to the OPD group). A one-to-one propensity score matching (56:56) was used to match the baseline characteristics of patients who underwent LPD and OPD. LPD was associated with significantly fewer clinically significant postoperative pancreatic fistulas (7.1% vs. 21.4%), fewer analgesic injections (*p* = 0.022), and longer operative time (321.8 vs. 268.5 min; *p* = 0.001) than OPD. There were no significant differences in 3-year overall and disease-free survival rates between the LPD and OPD groups. LPD had acceptable perioperative and oncological outcomes compared with OPD in elderly patients.

Tan et al. [18] retrospectively collected data from the three defined groups: LPD aged < 70 years (group I, 84 patients), LPD aged ≥ 70 years (group II, 56 patients) and OPD aged ≥ 70 years (group III, 28 patients). In elderly patients, when compared with OPD, LPD had the advantage of shorter time to start oral intake (*p* = 0.005) but the disadvantage of longer operative time (*p* < 0.001) and higher hospitalization cost (*p* < 0.001). There was no difference between the two groups in terms of postoperative stay, and proportion of reoperation, Clavien–Dindo classification, 30-day readmission and 90-day mortality.

More recently, Liang et al. [19] concluded from the small size of the groups studied (55 non-elderly patients (<70 years), 27 elderly patients (≥70 years) who underwent LPD, and 19 elderly patients who underwent OPD) that aging patients had higher overall morbidity than younger patients in LPD. However, LPD had several benefits such as reduced hospital stay and less intra-operative blood loss.

The aim of this study was to determine the outcomes of LPD in the elderly patients, by performing a systematic review and a meta-analysis of relevant studies.

## 2. Materials and Methods

### 2.1. Study Design

The methodological algorithm for this systematic review consisted of different steps, namely definition of search strategies, selection criteria, assessment of study quality, and relevant data abstraction [23]. Hence, The Preferred Reporting Items for Systematic Reviews and Meta-Analyses (PRISMA) statements checklist was used to structure our search methodology [24]. There was no need to obtain written consent from patients because no data of patients from our hospital were needed for the current review.

### 2.2. Study Inclusion and Exclusion Criteria

Comparative studies reporting the outcomes of LPD in elderly patients were analyzed. Two different comparative studies were conducted: first study compared outcomes between LPD vs. OPD in elderly patients; and the second study focused on the results after LPD in elderly versus non-elderly patients.

Case reports, reviews, non-comparative studies, commentaries and studies with fewer than 10 patients were not considered, neither were series published before 2012. Both benign and malignant tumor resections were included in the study. PICO framework was used to define the study selection criteria.

Primary outcomes were postoperative mortality rate at 30-day and 90-day mortality, and readmission rate. Secondary outcomes were complication rate (Clavien–Dindo, pancreatic fistula), complete resection (R0) rate, conversion rate and blood loss.

### 2.3. Literature Search Strategy

Online databases (Embase, Medline, Pubmed, Scopus, Cochrane) were screened. Specific key words were entered for each database to identify all eligible studies: (1) Pubmed: (“aged” [MeSH Terms] OR “aged” [All Fields] OR “aged patient” [All Fields] OR “aged people” [All Fields] OR “aged person” [All Fields] OR “aged subject” [All Fields] OR “elderly” [All Fields] OR “elderly patient” [All Fields] OR “elderly people” [All Fields] OR “elderly person” [All Fields] OR “elderly subject” [All Fields] OR “senior citizen” [All Fields] OR “senium” [All Fields]) AND (“pancreaticoduodenectomy” [MeSH Terms] OR “pancreaticoduodenectomy” [All Fields] OR “whipple operation” [All Fields] OR “brunschwig operation” [All Fields] OR “duodenopancreatectomy” [All Fields] OR “pancreatico duodenectomy” [All Fields] OR “pancreaticoduodenectomy” [All Fields] OR “pancreato duodenal resection” [All Fields] OR “pancreato duodenectomy” [All Fields] OR “pancreatoduodenal resection” [All Fields] OR “pancreatoduodenectomy” [All Fields] OR “total pancreatic duodenectomy” [All Fields] OR “whipple resection” [All Fields]) AND (“laparoscopic” [MeSH Terms] OR “laparoscopic” [All Fields] OR “laparoscopy” [All Fields] OR “Laparoscopies” [All Fields]) AND (“open” [All Fields]). (2) Embase: (‘aged’/exp OR ‘aged’ OR ‘aged patient’ OR ‘aged people’ OR ‘aged person’ OR ‘aged subject’ OR ‘elderly’ OR ‘elderly patient’ OR ‘elderly people’ OR ‘elderly person’ OR ‘elderly subject’ OR ‘senior citizen’ OR ‘senium’) AND (‘pancreaticoduodenectomy’/exp OR ‘whipple operation’ OR ‘brunschwig operation’ OR ‘duodenopancreatectomy’ OR ‘pancreatico duodenectomy’ OR ‘pancreaticoduodenectomy’ OR ‘pancreato duodenal resection’ OR ‘pancreato duodenectomy’ OR ‘pancreatoduodenal resection’ OR ‘pancreatoduodenectomy’ OR ‘total pancreatic duodenectomy’ OR ‘whipple resection’) AND (‘laparoscopy’/exp OR ‘laparoscopic’ OR ‘laparoscopies’) AND (‘open surgery’/exp OR ‘open surgery’ OR (‘open’ AND ‘surgery’)) (3) Scopus: (“aged” OR “aged patient” OR “aged people” OR “aged person” OR “aged subject” OR “elderly” OR “elderly patient” OR “elderly people” OR “elderly person” OR “elderly subject” OR “senior citizen” OR “senium”) AND (“pancreaticoduodenectomy” OR “whipple operation” OR “brunschwig operation” OR “duodenopancreatectomy” OR “pancreatico duodenectomy” OR “pancreaticoduodenectomy” OR “pancreato duodenal resection” OR “pancreato duodenectomy” OR “pancreatoduodenal resection” OR “pancreatoduodenectomy” OR “total pancreatic duodenectomy” OR “whipple resection”) AND (“laparoscopy” OR “laparoscopic” OR “Laparoscopies”) AND (“open surgery” OR “open surgery” OR (“open” AND “surgery”)) AND (LIMIT-TO (SUBJAREA, “MEDI”)) AND (LIMIT-TO (DOCTYPE, “ar”)) (4) Cochrane: (“aged” OR “aged patient” OR “aged people” OR “aged person” OR “aged subject” OR “elderly” OR “elderly patient” OR “elderly people” OR “elderly person” OR “elderly subject” OR “senior citizen” OR “senium”) AND (“pancreaticoduodenectomy” OR “whipple operation” OR “brunschwig operation” OR “duodenopancreatectomy” OR “pancreatico duodenectomy” OR “pancreaticoduodenectomy” OR “pancreato duodenal resection” OR “pancreato duodenectomy” OR “pancreatoduodenal resection” OR “pancreatoduodenectomy” OR “total pancreatic duodenectomy” OR “whipple resection”) AND (“laparoscopy” OR “laparoscopic” OR “Laparoscopies”) AND (“open surgery” OR “open surgery” OR (“open” AND “surgery”)).

### 2.4. Study Selection and Quality Assessment

The abstracts of the selected articles were independently and blindly screened for relevance by two reviewers (H.B. and I.K.). In order to achieve a high sensitivity and specificity, records were selected for further analysis if none of the reviewers rejected it. Data analyzing and extraction were performed by both reviewers independently.

Non-randomized studies were evaluated using the Newcastle–Ottawa Scale system.

Different conclusions or disagreements between the two reviewers were further discussed with the third and fourth reviewers (A.B. and D.B.).

### 2.5. Data Extraction and Analysis

Qualitative and quantitative analyses were performed based on results reported in included studies. The Mantel–Haenszel method was used to estimate the risk ratio (RR, 95%CI) for dichotomous outcome data; a RR of less than 1.00 was in favor of laparoscopy. Mean differences and 95%CI were used for continuous data. Outcome measures were extracted or calculated for each surgical approach (SD or median values). Heterogeneity was assessed with the I^2^ statistic.

The pooled estimates of the mean differences were calculated using a random effect model to take into account potential inter-study heterogeneity and to adopt a more conservative approach. The robustness of the results and the potential sources of heterogeneity were then explored using sensitivity analyses. The pooled effect was considered significant with *p* < 0.05.

## 3. Results

### 3.1. Literature Search and Selection

Figure 1 represents the PRISMA diagram followed by a selection of articles. Out of 1736 studies initially identified, 1601 were excluded after the screening of titles and abstracts because of irrelevance for this study. A full-text evaluation of the remaining 136 articles was performed. In the end, 5 studies were included in the meta-analysis, of which all 5 compared laparoscopic and open approaches for PD in elderly patients (Table 1), whereas 2 of them also compared the outcomes of laparoscopic surgery on younger and older patients (Table 2).

### 3.2. First Study: LPD vs. OPD in Elderly

All included studies were retrospectives. A total number of 2062 elderly patients who underwent LPD were analyzed. The elderly population was considered above 70 years in four studies (542) [18,19,21,22] and 75 in one study (1520) [20]. The most frequent pathology was adenocarcinoma.

Postoperative mortality rate at 30 days was reported in two studies [18,19]. The RR was 1.42, without significant differences (95%CI: 0.35–5.88; *p* = 0.62), although a moderate/high heterogeneity index was noted between studies (I^2^ = 69%) (Table 3a).

The analysis of readmission rate in hospital after surgery was reported in two studies [18,19] and showed no significant differences (RR = 1.67, 95%CI: 0.25–11.4; *p* = 0.6; I^2^ = 0%) (Table 3b).

The mortality rate at 90 days postoperatively was reported in three studies [18,19,20]. It occurred in 6.4% of LPD and 12% of OPD with a significant overall difference in favor of LPD (RR = 0.56; 95%CI: 0.32–0.96; *p* = 0.037, I^2^ = 0%) (Table 3c).

Clavien–Dindo complications were reported in three studies [17,18,21]. The RR for Clavien–Dindo I/II complications showed no significant differences (RR = 0.88, 95%CI: 0.33–2.34; *p* = 0.8) although a moderate heterogeneity index was noted between studies (I^2^ = 63.7%) (Table 3d). The Clavien–Dindo III-V complication rate was similar in both groups, with an overall RR of 0.57 (95%CI: 0.28–1.17; *p* = 0.12, I^2^ = 0%) (Table 3e).

The analysis of delayed gastric emptying was reported in three studies [18,19,21]. It occurred in 15.3 % of LPD and 30.8 % of OPD with a significant overall difference in favor of LPD (RR = 0.54; 95%CI: 0.33–0.87; *p* = 0.012, I^2^ = 0%) (Table 3f).

The analysis of surgical complication such as pancreatic fistula grade B/C was reported in four studies [18,19,21,22]. The postoperative pancreatic fistula rate was 16.7% in LPD and 23.8% in OPD without significant difference (RR = 0.61, 95%CI: 0.34–1.08; *p* = 0.09), low heterogeneity (I^2^ = 30%) (Table 3g).

Complete resection (R0) was reported in three studies [18,19,20] and was achieved in 82.1% by LPD approach and in 73.6% in open surgery, respectively. No significant difference was identified (RR = 1.27, 95%CI: 0.93–1.74; *p* = 0.13, I^2^ = 0%) (Table 3h).

Reoperation for complications was analyzed in three studies [18,19,21]. The RR for reoperation was slightly lower for LPD vs. OPD but no significant differences was reported (RR = 0.55, 95%CI: 0.21–1.45; *p* = 0.22, I^2^ = 0%) (Table 3i).

Intraoperative blood loss was reported in three studies [18,19,21]. Tee et al. [21] reported an average of 345 mL blood loss in LPD and 869 mL in OPD. Liang et al. [18] reported 100–400 mL blood loss in LPD and 200–700 mL in OPD. Tan et al. [18] reported similar quantities of blood loss: 200–500 mL in both groups. No significant difference was identified (RR = 1, 95%CI: 0.39–2.57; *p* = 0.996; I^2^ = 14.4%) (Table 3j).

### 3.3. Second Study: Outcome of LPD in the Elderly vs. Non-Elderly Patients

Two studies reported the comparison between elderly (≥70 years) and non-elderly patients (<70 years) [18,19], summarizing 93 elderly patients and 139 young patients.

The RR for the mortality rate at 90-day postoperatively was 2.3 in favor of non-elderly, without significant difference (RR = 2.3; 95%CI: 0.49–10.78; *p* = 0.29, I^2^ = 0%) (Table 4a).

The RR for readmission was similar between elderly and non-elderly (RR = 1.1; 95%CI: 0.1–10.82; *p* = 0.93), and a moderate heterogeneity index rate (I^2^ =60%) (Table 4b).

Clavien–Dindo III-V complications were less frequent in non-elderly patients (RR = 1.45; 95%CI: 0.39–5.35; *p* = 0.58) without significant results and a moderate heterogeneity (I^2^ = 63.3%) (Table 4c). The RR for Clavien–Dindo I/II complications showed no significant differences (RR = 0.69, 95%CI: 0.19–2.55; *p* = 0,58), although a moderate heterogeneity index was noted between studies (I^2^ = 66%) (Table 4d).

Rate of conversion was similar between the elderly and the non-elderly, with an overall RR of 1.03 (95%CI: 0.36–3; *p* = 0.95; I^2^ = 0%) (Table 4e).

The delayed gastric emptying occurred in 3.6% the elderly and 5.8% in the non-elderly, with an overall RR of 0.63 slightly in favor of the elderly (95%CI: 0.17–2.27; *p* = 0.48; I^2^ = 0%) (Table 4f).

The postoperative pancreatic fistula grade B/C incidence was similar in the elderly and non-elderly patients (RR = 0.99, 95%CI: 0.45–2.14; *p* = 0.97, I^2^ = 0%) (Table 4g).

Complete resection (R0) rate similar between the elderly and non-elderly, with an overall RR of 0.56 (95%CI: 0.1–3.15; *p*= 0.51; I^2^ = 0%) (Table 4h).

The RR for reoperation was slightly lower in the non-elderly vs. elderly patients (RR = 1.51; 95%CI: 0.35–6.54) without significant differences and lower heterogeneity (*p* = 0.58; I^2^ = 25%) (Table 4i).

The estimated blood loss was not significantly different between groups (RR = 1.36; 95%CI: 0.27–6.81; *p* = 0.16; I^2^ = 49%) (Table 4j).

The quality assessment of the included studies is represented in Table 5.

## 4. Discussion

In the past few decades, conventional surgery has shifted towards less invasive procedures, such as the laparoscopy approach to treat many pathologies [4]. Even more challenging from technical aspects and with a longer learning curve, the laparoscopic technique provides better outcomes regarding post-operative pain, decreases hospital length of stay and offers a quicker return to normal activity [4].

Population is aging worldwide, and the surgical needs of the elderly patients are increasing. The literature suggest that age itself might be an important risk factor for postoperative morbidity and mortality [25] as they faced multiple morbidities and geriatric syndromes and lower physiological reserve and preoperative nutritional conditioning than those of young patients [26,27,28,29]. Elderly patients benefit from a laparoscopic approach for many pathologies (cholecystectomy, hernia repairs, colorectal cancer resection, etc.), as it has been deemed safe and effective [4,30,31,32,33]. The typical benefits of laparoscopic surgery include reduced postoperative pain and period of hospital stay length, improved mobilization, faster return to normal activity, and fewer abdominal wall complications. Laparoscopy was followed by significantly lower rates of readmissions related to gastrointestinal, wound complications and malignancy [34]. For elderly patients with periampullary tumors, LPD might be an alternative surgical option [35].

On the other side, the laparoscopic procedure has some drawbacks, including prolonged operation time and impact of carbon dioxide pneumoperitoneum on circulatory and respiratory dynamics [35]. Although laparoscopy is minimally invasive in its dissection techniques, its physiology might lead to acidosis, produce changes in pulmonary mechanics, induce alteration in hemodynamic function and increase the risk of aspiration. These “physiological aspects” could be challenging in elderly patients with low cardiopulmonary reserve [4,35].

Some merits and limitations of LPD versus OPD are listed in Table 6.

After its first description in 1994 [6] LPD was not adopted with enthusiasm as it required a longer operative time and the need of perfect operational skills. At the beginning the proofs for obvious advantages after LPD were lacking [6,36]. Meanwhile, with the gradual deepening of the understanding of the anatomy of the pancreas and the development of new surgical instruments, LPD resection was adopted by more and more surgeons [37]. On larger cohorts of patients, LPD was proved to be equivalent to OPD regarding length of stay, margin-positive resection, lymph node count, and readmission rate [37,38]. There was a higher 30-day mortality rate with LPD, most likely related to the learning curve for the procedure [37,38]. Regarding the oncological outcomes, LPD was reported to be equivalent to OPD for margin status and lymph node removal on a cohort of 22,013 patients [39]. More recently, three RCTs documented that LPD was a safe and feasible procedure if performed by experienced teams [13,14,15]; LPD was associated with a significant shorter length of stay compared with OPD. However, for short-term morbidity and mortality rates, RCTs comparing LPD with OPD found conflicting results. Similar rates of 90-day mortality and of serious postoperative morbidities were noticed in two RCTs [13,15]. The RCTs conducted by van Hilst [16] was prematurely discontinued because of a difference in 90-day complication-related mortality in LPD group. In the RCT conducted by Poves et al. [14], patients with LPD experienced significantly lower rates of severe complication (Clavien–Dindo > 3) than those with OPD. A meta-analysis of three RCTs including 224 patients reported no significant benefit of laparoscopic approach compared to OPD in all aged categories [17]; no significant differences were noted regarding the 90-day mortality and perioperative morbidity. In this meta-analysis, the limitation included high risk of bias and moderate to very low certainty of evidence.

Taken together, the retrospective large studies and RCTs in all aged categories argue a minor clinical benefit of LPD compared with OPD despite extensive procedural expertise. For elderly patients, the efficacy and safety of LPD is still questionable. To date, only a few retrospective studies addressed the topic of LPD in elderly patients [18,19,20,21,22].

In this study, we identified and analyzed in a systematic manner data regarding the outcomes of LPD in elderly patients to offer clinicians and surgeons evidence for daily practice. The meta-analyses of previous relevant studies on this topic were recently addressed in two meta-analyses [40,41]. A comprehensive recent meta-analysis summarizing 34 studies involving 46,729 patients compared the outcomes of LPD vs. OPD and provided a subgroup analysis on elderly patients (≥70 years) [42].

The most important finding of our study was the significant lower relative risk for 90-day mortality rate in elderly patients that benefited from laparoscopic approach compared to classical surgery (RR = 0.56; 95%CI: 0.32–0.96; *p* = 0.037). Our results are different from those reported in all aged categories [17] and in the meta-analyses conducted by Wang et al. [40] and by Zhang et al. [41] focusing on elderly patients, that report similar odds ratio for 90-day mortality rate between OPD and LPD. These differences might be partially explained by different measurement tools and included studies. Our study and the meta-analysis performed by Zhang et al. [41] reported the risk ratio; the meta-analysis conducted by Wang et al. [40] reported the odds ratio. The potential inter-study heterogeneity of our analysis for 90-day mortality was 0%, similar to that reported by Wang et al. [40] at 1%; however, Zhang et al. [41] reported higher heterogeneity among included studies, which was at 42%. The meta-analysis conducted by Wang et al. [40] included one small size study with octogenarians [43], without any death recorded.

In our meta-analysis, even the mortality rate at 90 days postoperatively was higher in elderly patients receiving LPD, the difference was not statistically significant (RR = 2.3; 95%CI: 0.49–10.78; *p* = 0.29, I^2^ = 0%) compared with non-elderly LPD; similar results were observed in the meta-analysis conducted by Wang et al. [40]. However, in the meta-analysis conducted by Zhang et al. [41] the 90-day mortality rate was statistically significant different, but the heterogeneity of the included studies was higher (RR = 2.29; 95%CI: 1.14–4.6, *p* = 0.02, I^2^ = 0%).

Regarding the mortality rate at 30 days, our analysis did not find a significant difference between open or laparoscopic approach for PD in elderly patients (RR = 1.42, 95%CI: 0.35–5.88; *p* = 0.62). Previous studies reported a higher 30-day mortality rate for LPD in all aged-category patients [37,38], most likely related to learning curve. Our results document that elderly patients did not have higher 30-day mortality rates in case of laparoscopic approach. Similar results were observed in the meta-analyses focusing on elderly patients [40,41]. In the largest studied included in our analyses, Chapman et al. [20] reported a 30-day mortality of 8% for the laparoscopic approach, compared with 3.8% in open procedures, but the difference was not statistically significant (*p* = 0.26). Tee et al. [21] also reported higher 30-day mortality rates in elderly patients receiving LPD, but without significance (4.4% vs. 1.3%).

We did not notice any significant difference regarding the secondary outcomes (complications, pancreatic fistula, blood loss, gastric emptying time) between LPD and OPD in elderly patients. This result highlights that in elderly patients with comorbidities, the laparoscopic approach did not significantly unbalance their acid-basic, hemodynamic or respiratory homeostasis. The meta-analyses conducted by Zhang et al. [41] and by Wang et al. [40] reported more optimistic results for the elderly: pancreas fistula and delayed gastric emptying in the LPD group were significantly lower than those in the OPD group. In the subgroup of elderly patients reported by Liu et al. [42], significantly lower rates were observed for delayed gastric emptying time.

Readmission rate in hospital after surgery was similar in elderly patients comparing LPD with OPD; also, no significant differences were noted for readmission rate in elderly vs. non-elderly LPD patients. Our findings were similar with previous reports [40,41].

The complete resection (R0) rate was similar in elderly patients from LPD group compared to OPD; moreover, no difference was found when the complete resection rate was compared in elderly vs. non-elderly patients with LPD. In oncological terms, LPD in elderly patients offers the same curative rates as in younger patients. The same results were reported in previously published meta-analysis [40,41].

Two included studies [18,19] compared the outcomes between elderly (≥70 years) (summarized 93 cases) and non-elderly patients (<70 years) (139 non-elderly) submitted to LPD. Tan et al. [18] investigated similar cohorts regarding pathologic diagnosis, harvest lymph nodes and the proportion of R0 resection cases. Even the tumor size of the elderly group was higher than the other (2.5 cm vs. 2.2 cm, *p* = 0.068), the authors did not find differences between elderly and no-elderly LPD patients in terms of operative time, conversion rate, all drainage tube removal time, time of postoperative stay, and proportion of vascular reconstruction, reoperation, pancreatic fistula, delayed gastric emptying, hemorrhage, Clavien–Dindo classification, 30-day readmission and 90-day mortality. The elderly LPD patients had a higher blood loss (*p* = 0.003), and higher proportion of intraoperative transfusion (17.9% versus 6.0%, *p* = 0.026) in the study of Tan et al. [18]. In the study of Liang et al. [19], in elderly people, with significant more baseline morbidity (higher ASA score and larger proportion of malignancy than younger patients), the laparoscopic approach had similar 90-mortality (2% vs. 7%, *p* = 0.26). The estimated blood loss was similar between groups of the elderly vs. the non-elderly in the study by Liang et al. [19]. Although initially Liang et al. [19] found significant morbidity rates in elderly LPD patients, our results from a larger cohort report similar rates of morbidity to non-elderly patients. No significant differences were noted regarding rate of conversion, complete resection rate, Clavien–Dindo complications, postoperative pancreatic fistula, and readmission rate. Additionally, the estimated blood loss was not significantly different in elderly vs. non-elderly patients receiving LPD (RR = 1.36; 95%CI: 0.27–6.81; *p* = 0.16; I^2^ = 49%) when all the patients were combined in the meta-analysis. The same results were observed in the studies conducted by Wang et al. [40] and by Zhang et al. [41].

Most of the included studies did not report long-term follow-up results. As per one study, Chapman et al. [20] found that LPD had a longer median survival time in older people than OPD.

In the last decade, advances in the surgical techniques and perioperative management have evolved towards a robotic approach for PD. The limitation of laparoscopic surgery (two-dimensional imaging, restricted range of motion of the instruments, and poor ergonomic positioning of the surgeon) were exceeded by the robotic PD (improved visualization and greater dexterity). Recent studies conducted by Liu et al. [44] and Paolini et al. [45] compared the outcomes of robotic PD (169) vs. open approach (133) in elderly patients. The robotic PD group had a significant shorter hospitalization period than OPD group. The authors reported no significant differences in the rates of pancreatic fistula, bile leakage, delayed gastric emptying, major morbidity, reoperation, or 90-day mortality between the two groups [43,44].

Limitations of our study are related to the retrospective nature of analyzed studies (selection bias) and limited number of included patients. Our searched databases were limited to Embase, Medline, Pubmed, Scopus and Cochrane; the database Web of Science was not screened, and it is likely that some relevant studies were missed out as part of the critical analysis. Another possible bias might be represented by data from institutions performing 1–4 cases of LPD per year; in these centers, healthier patients, more physically fit, with fewer comorbidities, and less advanced cancers might have been submitted to LPD. As most studies lacked long-term follow-up data, we cannot have definitive results. Larger sample size multicenter studies with longer follow-up time are required to be carried out to confirm these data.

## 5. Conclusions

Our meta-analysis argues for laparoscopic approach for PD in elderly patients (more than 70 years old), as 90-day mortality rates are significantly lower compared with the open approach. The laparoscopic approach has similar mortality rate at 30-day, readmission rate in hospital, Clavien–Dindo complications, pancreatic fistula grade B/C, complete resection rate, reoperation for complications and blood loss as an open approach. In the laparoscopic approach, comparing elderly vs. younger patients, no significant differences were seen regarding mortality rate at 90 days, readmission rate in hospital, Clavien–Dindo complications, conversion rate, delayed gastric emptying, pancreatic fistula grade B/C, complete resection, reoperation for complications, blood loss. As this meta-analysis was based on retrospective studies, larger prospective studies are required to further validate our observations.

## Figures and Tables

**Figure 1 life-12-01810-f001:**
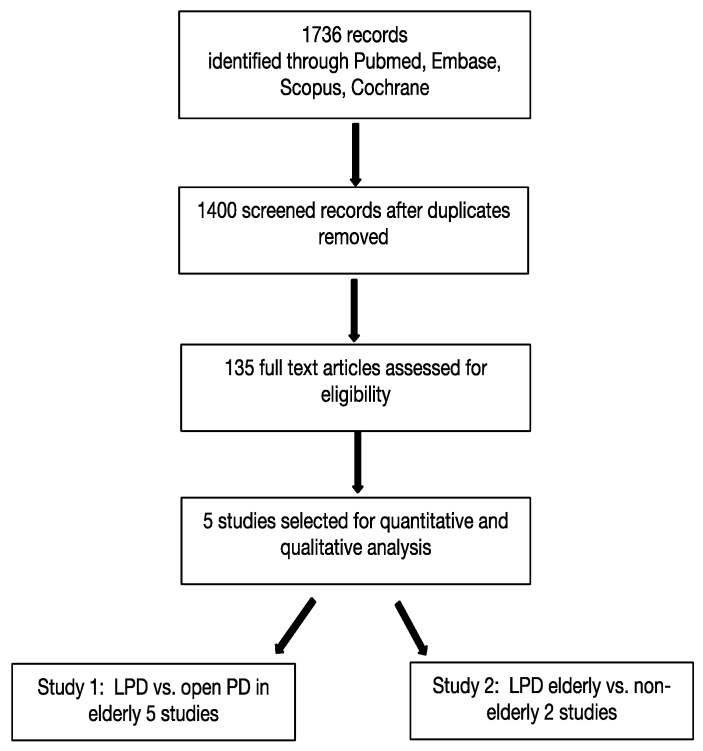
Prisma flowchart.

**Table 1 life-12-01810-t001:** LPD vs. OPD in elderly.

Source	Period	Type of Study	Nr of Pt	Age of Cut Off	Approach	Median Age	ASA Score Class or Mean ± SD	Type of Pathology	Tumour Size (cm, IQR or ±SD)
Tan [18]	2015–2019	R	56/28	≥0	LPD OPD	75.2 (±4.4) 74.7 (±4.6)	II 37/19 III 19/9	PC 21/11 CC 31/4 AC 2/3 DC 8/4	2.5 (1.9–4.3) 3.3 (2.0–4.0)
Liang [19]	2015–2018	R	27/19	≥70	LPD OPD	74 (±4) 76 (±5)	II 19/8 III 18/8	PC 12/15 AC 12/2	2.6 (±1) 3.1 (±1)
Chapman [20]	2010–2013	R	248/1520	≥75	LPD OPD	79.6 (±3.5) 79.5 (±3.4)	N/A	PC 248/1520	<2: 25/125 2–4: 169/1002 >4: 49/329
Tee [21]	2007–2014	R	113/225	≥70	LPD OPD	76.5 (±4.3) 76.4 (±4.5)	I-II 30/67 III-IV 83/158	PC 53/121 CC 4/15 AC 9/33 DC 2/11	N/A
Shin [22]	2014–2017	R	56/270	≥70	LPD OPD	74.8 (±3.7) 74.6 (±3.5)	2.1± 0.5 2.1±0.4	PC 14/115 CC 19/92	N/A

R: retrospective; LPD: laparoscopic pancreatoduodenectomy; OPD: open pancreatoduodenectomy; N/A: not assessed.

**Table 2 life-12-01810-t002:** Elderly vs. young LPD.

Source	Period	Type of Study	Nr of Pt	Age of Cut Off	Median Age	Type of Pathology	ASA Score Class or Mean ± SD	Tumor Size (cm, IQR or ±SD)
Tan [18]	2015–2019	R	56/84	≥70	75.2 (±4.4) 60.7 (±7.5)	PC 21/22 CC 31/20 AC 2/1 DC 8/13	II 37/66 III 19/18	2.5 (1.9–4.3) 2.2 (1.7–3.0)
Liang [19]	2015–2018	R	27/55	≥70	74 (±4) 59 (±9)	PC 12/18 AC 12/17	II 19/42 III 8/1	2.6 (±1) 2.8 (±1.4)

**Table 3 life-12-01810-t003:** Studies evaluating LPD vs. OPD in elderly. (**a**) postoperative mortality rate at 30-day; (**b**) readmission rate in hospital; (**c**) mortality rate at 90-day; (**d**) Clavien–Dindo I/II complications; (**e**) Clavien–Dindo III-V complications; (**f**) gastric emptying; (**g**) pancreatic fistula grade B/C; (**h**) complete resection; (**i**) reoperation for complications; (**j**) blood loss.

(**a**)	Postoperative mortality rate at 30 days	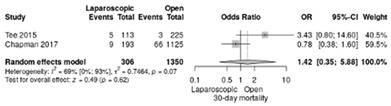
(**b**)	Readmission rate in hospital	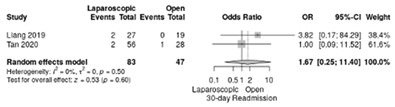
(**c**)	Mortality rate at 90 days	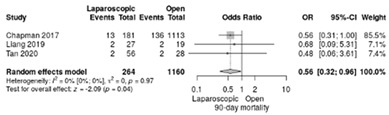
(**d**)	Clavien–Dindo I/II complications	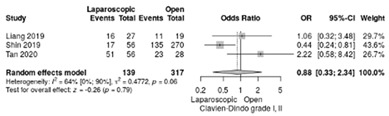
(**e**)	Clavien–Dindo III-V complications	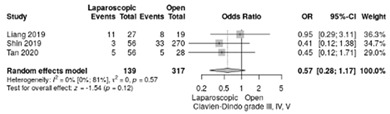
(**f**)	Delayed gastric emptying	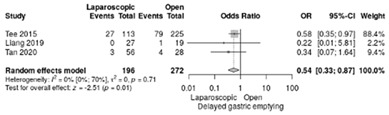
(**g**)	Pancreatic fistula grade B/C	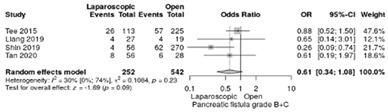
(**h**)	Complete resection (R0)	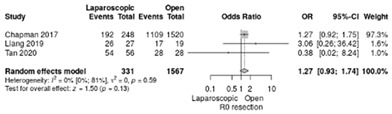
(**i**)	Reoperation for complications	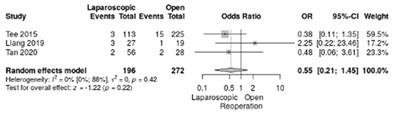
(**j**)	Blood loss	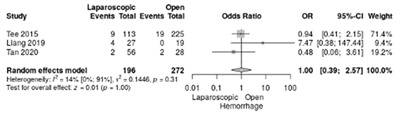

**Table 4 life-12-01810-t004:** Studies evaluating LPD in elderly vs. non-elderly. (**a**) mortality rate at 90 days; (**b**) readmission rate in hospital; (**c**) Clavien–Dindo III-V complications; (**d**) Clavien–Dindo I/II complications; (**e**) conversion rate; (**f**) delayed gastric emptying; (**g**) pancreatic fistula grade B/C; (**h**) complete resection; (**i**) reoperation for complications; (**j**) blood loss.

(**a**)	Mortality rate at 90 days	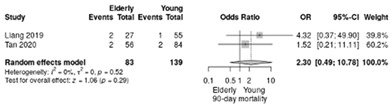
(**b**)	Readmission rate in hospital	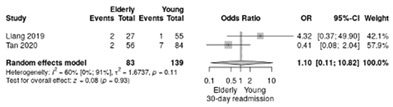
(**c**)	Clavien–Dindo III-V complications	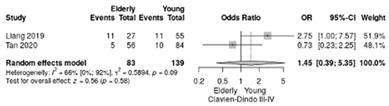
(**d**)	Clavien–Dindo I/II complications	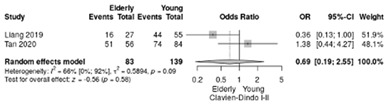
(**e**)	Conversion rate	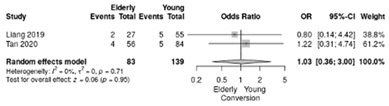
(**f**)	Delayed gastric emptying	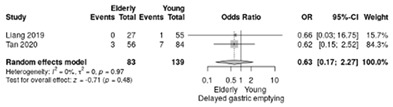
(**g**)	Pancreatic fistula grade B/C	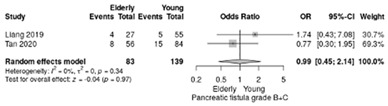
(**h**)	Complete resection (R0)	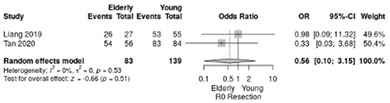
(**i**)	Reoperation for complications	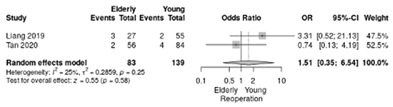
(**j**)	Blood loss	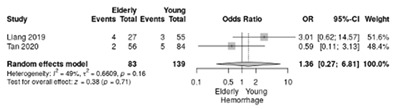

**Table 5 life-12-01810-t005:** The quality assessment of the included studies [18,19,20,21,22].

Article	Selection				Comparability	Outcome			Score (Risk of Bias)
	1	2	3	4	5	6	7	8	
Tan et al.	☆	☆	☆	/	☆	☆	☆	☆	7 (low)
Liang et al.	☆	☆	☆	/	☆	☆	☆	☆	7 (low)
Chapman et al.	☆	☆	☆	/	☆	☆	☆	☆	7 (low)
Tee et al.	☆	☆	☆	/	☆	☆	☆	☆	7 (low)
Shin et al.	☆	☆	☆	/	☆	☆	☆	☆	7 (low)

**Table 6 life-12-01810-t006:** Laparoscopic vs. open surgical approach: benefits and disadvantages [4,30,31,32,33,34,35].

	Benefits	Disadvantages
Laparoscopic approach	Reduced postoperative pain Decreased hospital length of stay Improved mobilization Faster return to normal activity Fewer abdominal wall complications Lower rates of readmissions related to gastrointestinal, wound complications and malignancy and subsequently lower costs	More challenging based on technical aspects, with a longer learning curve for surgeons experienced with open approach Carbon dioxide pneumoperitoneum used in laparoscopic approach might lead to acidosis, produce changes in pulmonary mechanics, induce alteration in hemodynamic function and increase the risk of aspiration
Open surgery	Standardized procedures Established training	More blood loss compared with laparoscopy Increased postoperative pain, considering the large incision of abdominal wall Increased hospital length of stay Abdominal wall complications Increased recovery time compared with laparoscopy

## Data Availability

Not applicable.

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
