# Peer review of "Laparoscopic Pancreatoduodenectomy in Elderly Patients: A Systematic Review and Meta-Analysis"

_life, 2022, doi:10.3390/life12111810_

Round 1
Reviewer 1 Report
Bartos et al. present a systematic review comparing minimal-invasive vs open PD in elderly patients and differenced between younger and elderly patients undergoing minimal-invasive PD. The rationale to perform this study is given. The methods are adequate. The results are well presented. The discussion is balanced.
I have some suggestions:
Is it III-V and not III/V in Fig 2e/ 3c?
It is Clavien-Dindo not Dingo (https://www.proctomed.ch/pd-dr-med-daniel-dindo/)
Please give the reader an overview what is known about minimal-invasive PD by citing the “Evidence Map of Pancreatic Surgery” (Evidence Map of Pancreatic Surgery-A living systematic review with meta-analyses by the International Study Group of Pancreatic Surgery (ISGPS). Surgery. 2021 Nov;170(5):1517-1524.) You can access it here www.emps.evidencemap.surgery and find all current and ongoing trials.
Please also compare the results of the studies you included to the benchmarks in the summary of the Evidence Map in your discussion section.
There is a gold standard for searches in surgical reviews, please site it: Systematic reviews in surgery-recommendations from the Study Center of the German Society of Surgery. Langenbecks Arch Surg. 2021 Sep;406(6):1723-1731. If you search for non-randomised studies, the “web of science” should be one of the databases, please add this database to your search or discuss it as a limitation.
Reviewer 2 Report
Please find comments attached

Round 2
Reviewer 2 Report
Some minor changes requested. Please find attached.

Author Response
Dear Editors and Reviewer,
We are very grateful for the comments of the reviewers. Based on these observations, we revised our manuscript, and we are hoping that our answers will increase its scientific value. The corrections are highlighted in red.
Reviewer 2
- We correct the English language according with the suggestions.
- We added the suggested references for nutritional status in elderly patients.
- We modified the content and added references to Table I.
- We deleted the paragraph, which referred to laparoscopic approach in elderly patients with other pathologies.
- In the Introduction section we did not delete the paragraph discussing the Evidence Map of Pancreatic Surgery, as it was requested by Reviewer 1 in the first round of corrections.
Best regards,
Lidia Ciobanu